# Hepatic Insulin Resistance in Hyperthyroid Rat Liver: Vitamin E Supplementation Highlights a Possible Role of ROS

**DOI:** 10.3390/antiox11071295

**Published:** 2022-06-29

**Authors:** Gianluca Fasciolo, Gaetana Napolitano, Marianna Aprile, Simona Cataldi, Valerio Costa, Alfredo Ciccodicola, Sergio Di Meo, Paola Venditti

**Affiliations:** 1Dipartimento di Biologia, Università di Napoli Federico II, 80126 Naples, Italy; gianluca.fasciolo@unina.it (G.F.); sergio.dimeo@unina.it (S.D.M.); 2Dipartimento di Scienze e Tecnologie, Università degli Studi di Napoli Parthenope, Centro Direzionale, Isola C4, 80143 Naples, Italy; gaetana.napolitano@uniparthenope.it (G.N.); alfredo.ciccodicola@igb.cnr.it (A.C.); 3Institute of Genetics and Biophysics Adriano Buzzati Traverso, National Research Council, Pietro Castellino Street 111, 80131 Naples, Italy; marianna.aprile@igb.cnr.it (M.A.); simona.cataldi@igb.cnr.it (S.C.); valerio.costa@igb.cnr.it (V.C.)

**Keywords:** oxidative damage, ROS production, antioxidant enzymes, Akt, JNK, HOMA index, *Slc2a1*, *Slc2a2*, *Ppara*, *Pparg*, *Cd36*, *Irs2*, *Ilb*

## Abstract

Thyroid hormones are normally involved in glycaemic control, but their excess can lead to altered glucose metabolism and insulin resistance (IR). Since hyperthyroidism-linked increase in ROS results in tissue oxidative stress that is considered a hallmark of conditions leading to IR, it is conceivable a role of ROS in the onset of IR in hyperthyroidism. To verify this hypothesis, we evaluated the effects of vitamin E on thyroid hormone-induced oxidative damage, insulin resistance, and on gene expression of key molecules involved in IR in the rat liver. The factors involved in oxidative damage, namely the total content of ROS, the mitochondrial production of ROS, the activity of antioxidant enzymes, the in vitro susceptibility to oxidative stress, have been correlated to insulin resistance indices, such as insulin activation of hepatic Akt and plasma level of glucose, insulin and HOMA index. Our results indicate that increased levels of oxidative damage ROS content and production and susceptibility to oxidative damage, parallel increased fasting plasma level of glucose and insulin, reduced activation of Akt and increased activation of JNK. This last result suggests a role for JNK in the insulin resistance induced by hyperthyroidism. Furthermore, the variation of the genes *Pparg*, *Ppara*, *Cd36* and *Slc2a2* could explain, at least in part, the observed metabolic phenotypes.

## 1. Introduction

Although the regulation of glucose metabolism largely depends on insulin and glucagon, thyroid hormones also contribute to glucose homeostasis. The examination of the effects of insulin and thyroid hormones on body tissues shows that thyroid hormones act as insulin agonists or antagonists [1] with finely balanced actions that contribute to glycemic control. Conversely, altered thyroid hormone levels can alter the glycemic balance, leading to impaired carbohydrate metabolism. More than 100 years ago, evidence was found that an alteration of glucose metabolism develops in hyperthyroidism and can induce a condition named thyroid diabetes [2].

The liver exerts a key-role in the development of thyroid diabetes. Indeed, hepatic insulin resistance (IR) is the main condition associated with non-insulin-dependent diabetes mellitus which can impair insulin sensitivity in other insulin-dependent tissues such as muscle and adipose tissue [3,4], which absorb glucose from plasma during the postprandial phase.

The thyroid hormone exerts extensive effects on the expression of genes that influence a wide range of pathways that modulate various cellular functions. Therefore, it has been hypothesized that the action of thyroid hormones on the expression of specific genes may provide an explanation for the development of insulin resistance in liver tissue. In fact, the thyroid hormone increases the expression of the gene of glucose-6-phosphatase, the enzyme that hydrolyses glucose-6-phosphate and is involved in the final step in gluconeogenesis and glycogenolysis, thus playing a key role in the homeostatic regulation of blood glucose levels [5]. Moreover, the thyroid hormones positively regulate the enzymes: phosphoenolpyruvate carboxykinase (PEPCK), catalysing the rate-controlling step of gluconeogenesis [6]; pyruvate carboxylase, involved in the synthesis of oxaloacetate from pyruvate [7]; and the expression of the glucose transporter 2 (GLUT 2) located on the plasma membranes of hepatocytes [8]. In this way, thyroid hormones increase gluconeogenesis and glycogenolysis [9] and plasma glucose levels [10].

Although the effects above-mentioned can contribute to altered glucose homeostasis occurring in hyperthyroid tissues, it has recently been suggested that IR can depend on the thyroid hormone-induced activation of the synthesis of components of the mitochondrial electron transport chain (ETC). Some of these components are responsible for the electron leak that leads to the production of superoxide anion and other reactive oxygen species (ROS) [11,12]. ROS, which can damage cellular components compromising their function, have been implicated in several disorders, including insulin resistance [13].

The mechanisms by which ROS impair insulin signaling are not fully characterized. The ROS effects on IR have been ascribed to alterations in various intracellular signaling pathways, including serine/threonine kinases c-jun NH2 terminal kinases (JNK) [14] that are involved in insulin resistance and obesity in rats [15]. Indeed, JNK can phosphorylate both the insulin receptor [16] and insulin-receptor-substrate (IRS) [17,18] on serine/threonine residues, thus inhibiting the physiological pathway of insulin through steric hindrance.

If the onset of insulin resistance in hyperthyroidism depends on oxidative stress due to increased ROS production, the restoration of redox homeostasis could effectively prevent it. To this end, in the present work, we investigated the ability of vitamin E supplementation to reduce the marker levels of oxidative stress and ROS production and content, in experimentally rendered hyperthyroid rats. We also determined whether the antioxidant protection prevents the reduction of insulin sensitivity in the liver by evaluating the activation of the signal activated downstream by the insulin binding to its receptor. Particularly, we analysed the serine/threonine kinase Akt, which in the liver promotes the storage of glucose as glycogen, inhibits the breakdown of glycogen, stimulates the conversion of excess glucose into lipids, and inhibits de novo production of glucose [19]. Moreover, we evaluated the effects of thyroid hormone and vitamin E on the activation of JNK, the expression levels of genes encoding glucose transporters (e.g., *Slc2a1* and *Slc2a2*), the factors involved in hepatic glucose/lipid homeostasis and insulin signalling (e.g., *Ppara*, *Pparg*, *Cd36*, *Irs2*), as well as *Il1b* which is strongly related to insulin resistance [20,21,22].

Our results show that antioxidant supplementation may attenuate hepatic insulin resistance and suggest that hyperthyroidism-linked oxidative stress may at least partially be responsible for the onset of this condition in the liver.

## 2. Materials and Methods

### 2.1. Animals

The experiments were conducted on 90-day-old male Wistar rats (Envigo, Italy). Up to the age of 80 days, the animals were given the same control diet, a commercial rat food purchased from Mucedola (Milan, Italy), containing 70 mg/kg of vitamin E (α-tocopherol). From day 80 to day 90, half of the 32 animals recruited for the study received, once a day, i.p. injection of triiodothyronine (T_3_) (Sigma-Aldrich, St. Louis, MO, USA) (50 μg/100 g body weight) [23]. In addition, half of control and hyperthyroid animals received a vitamin E supplemented diet consisting of commercial rat food to which α-tocopherol (Sigma-Aldrich) was added at a final concentration of 700 mg/kg. The other rats received the control diet. It has been found that this supplementation dose can reduce oxidative damage in sedentary and trained rats [24] and attenuate the changes induced by thyroid hormone on cardiac electrical activity in rats [25].

Thus, there were four groups of rats: euthyroid (E), euthyroid treated with vitamin E (E + VE), hyperthyroid (H), and hyperthyroid treated with vitamin E (H + VE). All animals were kept in the same environmental conditions, one per cage, at a room temperature of 24 ± 1 °C, with a constant artificial circadian cycle of 12 h of light and 12 h of darkness, and 50 ± 10% relative humidity, and received water and food on an ad libitum basis.

At the end of the treatment period, the animals were anaesthetized by intraperitoneal injection of Zoletil 60 mg/kg b.w. Arterial blood samples were collected and subsequently analysed to determine plasma levels of free triiodothyronine (FT_3_). Hearts and livers were quickly excised and placed in an ice-cold homogenizing medium (HM) (220 mM mannitol, 70 mM sucrose, 1 mM EDTA, 0.1% fatty acid-free albumin, 10 mM Tris, pH 7.4). Hearts were weighed after great vessels and valves were removed and ventricles and atria were flushed of blood.

The treatment of animals in these experiments was in accordance with the guidelines set forth by the University’s Animal Care Review Committee and received the authorization (code 836/2019 PR) from the Italian Ministry of Health.

### 2.2. Tissue Preparation

The livers were weighed, finely chopped, washed and gently homogenized in HM using a Potter-Elvehjem glass homogenizer set at a standard velocity (500 r.p.m.) for 1 min. Aliquots of 1:5 (weight/volume) homogenates were used for analytical procedures and the preparation of mitochondrial fractions.

Circulating levels of FT_3_ and FT_4_ were evaluated on plasma samples, using a ELISA kits (FT_3_ and FT_4_ Elisa kit MyBioSource, San Diego, CA, USA).

### 2.3. Mitochondria Preparation

The homogenates were freed from debris and nuclei by centrifugation at 500× *g* for 10 min at 4 °C. The resulting liver supernatants were centrifuged at 10,000× *g* for 10 min at 4 °C. The mitochondrial pellets were resuspended in wash buffer (WB) (220 mM mannitol, 70 mM sucrose, 1 mM EGTA, 20 mM Tris, (pH 7.4) and centrifuged at the same sedimentation velocity. The mitochondrial preparations were washed in this way twice before the final suspension in WB.

The protein content in mitochondrial preparations was determined by the biuret method, with bovine serum albumin as standard.

### 2.4. Tissue Content of Vitamin E

Tissue aliquots were deproteinized with methanol and extracted with n-hexane to determine the vitamin E content. The extracts were evaporated under N_2_ at 40 °C, and the residues were dissolved in ethanol. Vitamin E content was determined using the HPLC procedure of Lang [26]. Quantification was achieved using an external standard.

### 2.5. Redox State Evaluation

The extent of lipid peroxidative processes in homogenates and hepatic mitochondria was assessed by measuring the levels of lipid hydroperoxides (Hps) in 10 µg of homogenate or 0.1 mg of mitochondrial proteins diluted in 0.1 M monobasic phosphate buffer, pH 7.4 [27]. The hydroperoxides content was determined by measuring the stoichiometrically correlated reduction in NADPH absorbance, at 340 nm, due to the coupled reactions catalysed by the enzyme glutathione peroxidase and glutathione reductase in the presence of GSH. Levels of lipid hydroperoxides were expressed as oxidized nmol NADPH∙min^−1^∙mg^−1^ protein.

The extent of oxidative damage to proteins was determined by measuring the level of protein-bound carbonyls (CO). CO levels were revealed by the reaction of CO with 2,4-dinitrophenyl hydrazine (DNPH) using a simplified method adapted to a standard 96-well plate [28]. Briefly, 10 µL of homogenate previously treated with a lysis buffer (100 mM NaH_2_PO_4_, 0.2% digitonin, PMSF 80 µg/mL, leupeptin 10 µg/mL, pepstatin 14 µg/mL, aprotinin 10 µg/mL, pH 7.4) and 10% streptomycin was incubated with an equal volume of 10 mM DNPH (in 2.5 N HCl) at room temperature for 10 min. Subsequently, 5 µL of sodium hydroxide (6 N) was added to the mixture and incubated again for 10 min at room temperature. Immediately, the absorbance at 450 nm was recorded in a multimode microplate reader (Synergy HT, Biotek, Winooski, VT, USA). Protein carbonyls were calculated using the DNPH extinction coefficient at 450 nm (ε = 22,308 M^−1^ cm^−1^) and an optical path length of 0.1 cm and expressed as µmol of carbonyls/mg of protein. Total homogenate proteins were evaluated using the Bradford reagent according to the manufacturer’s instructions (Bio-Rad, Hercules, CA, USA).

### 2.6. Susceptibility to Oxidative Stress

The susceptibility of hepatic homogenates to oxidative stress in vitro was evaluated by the variation in hydroperoxide levels induced by treatment of 10% of tissue homogenate with Fe and ascorbate (Fe/As), at a concentration of 100/1000 μM, for 10 min at room temperature. The reaction was stopped by adding 0.2% 2.6-di-t-butyl-p-cresol (BHT) and the hydroperoxide levels were evaluated as previously described.

### 2.7. Reactive Oxygen Species Determination

ROS content was measured after ROS-induced conversion of 2′,7′-dichlorodihydrofluorescin diacetate (DCFH-DA, nonfluorescent compound) in dichlorofluorescein (DCF, fluorescent compound) according to Driver [29]. Briefly, 12.5 µg of homogenate proteins in 200 μL of monobasic phosphate buffer 0.1 M, pH 7.4, were incubated for 15 min with 10 µM DCFH-DA. Then, 100 µM FeCl_3_ was added, and the mixture was incubated for 30 min. The conversion of DCFH-DA to the fluorescent product DCF was measured using a multimode microplate reader (Synergy™ HTX Multimode Microplate Reader, BioTek, Winooski, VT, USA) with excitation and emission wavelengths of 485 and 530 nm, respectively. Background fluorescence (conversion of DCFH to DCF in the absence of homogenate) was corrected with a blank read in parallel.

### 2.8. NAD(P)H Oxidase (NOX) Activity Assay

The NOX activity was evaluated with a spectrophotometric method (550 nm) according to Suzuki, with modifications [30,31]. The measurements were performed using 0.2 mg of homogenate proteins following the reduction of ferricytochrome c acetylated (80 μM) in the presence of NADPH at room temperature. The NOX activity was calculated from the difference of the measurements performed in the presence and absence of 100 μg/mL of superoxide dismutase and expressed as µmol cytochrome c reduced∙min^−1^∙mg^−1^ protein.

### 2.9. H_2_O_2_ Mitochondrial Release

The mitochondrial rate of H_2_O_2_ release was determined using a computerized fluorometer (JASCO Deutschland GmbH, Pfungstadt, Germany) equipped with a cell thermostated at 30 °C. The increase in fluorescence (excitation at 320 nm, emission at 400 nm) is linked to the oxidation of p-hydroxyphenylacetate (PHPA) in the presence of H_2_O_2_, catalyzed by horseradish peroxidase (HRP) [32]. In our experiments 0.1 mg∙mL^−1^ of mitochondrial proteins were incubated with the reaction solution (HRP 6 U/mL, PHPA 200 μg/mL, KCl 145 mM, Hepes 30 mM, KH_2_PO_4_ 5 mM, MgCl_2_ 3 mM, EGTA 0.1 mM, 1% BSA, pH 7.4).

The reaction was induced by the addition of complex I-bound respiratory substrates, 10 mM pyruvate plus 2.5 mM malate. To obtain information on H_2_O_2_ release during state 3 of mitochondrial respiration, ADP at 500 μM concentration was added.

### 2.10. Activities of Antioxidant Enzymes (GPX, GR, SOD, CAT)

The activity of the enzyme glutathione peroxidase (GPX) was measured at 37 °C using H_2_O_2_ as substrate in the presence of GSH and following the rate of oxidation of NADPH in the reaction, catalyzed by glutathione reductase, to reduce the obtained GSSG [33].

The activity of the enzyme glutathione reductase (GR) was measured at 37 °C using GSSG as a substrate and following the oxidation rate of NADPH [34].

For both procedures, the oxidation rate of NADPH was measured using a multi-mode microplate reader (Synergy™ HTX Multi-Mode Microplate Reader, BioTek).

The activity of catalase was determined with the method of Aebi [35].

The specific activity of superoxide dismutase was measured spectrophotometrically at 25 °C, by monitoring the decrease in the reduction rate of cytochrome c at 550 nm induced by superoxide radicals, generated by the xanthine–xanthine oxidase system. Liver homogenates were incubated in a medium containing 0.1 mM EDTA, 2 mM KCN, 50 mM KH_2_PO_4_, pH 7.8, 20 mM cytochrome c, 5 mM xanthine, and 0.01 U of xanthine oxidase. A unit of SOD activity is defined as the concentration of the enzyme that inhibits the reduction of cytochrome c by 50% in the presence of xanthine + xanthine oxidase [36].

### 2.11. Oxygen Consumption

Oxygen consumption rates in tissue homogenates and mitochondria were determined at 30° using an Hansatech respirometer in 1.0 mL of incubation medium (145 mM KCl, 30 mM Hepes, 5 mM KH_2_PO_4_, 3 mM MgCl_2_, 0.1 mM EGTA, 1% BSA, pH 7.4) with 50 μL of 20% (*w/v*) homogenate or 0.25 mg of mitochondrial protein per mL. Pyruvate plus malate (10 and 2.5 mM, respectively) were used as respiratory substrates, in the absence (State 4) and in the presence (State 3) of 500 μM ADP. State 4 oxygen consumption of homogenates was measured by adding 2 μg/mL of oligomycin [37].

### 2.12. Basal Glucose and Insulin Serum Level

To obtain the basal glucose and insulin levels, the rats fasted for 12 h. Blood samples were taken from a cut in the tail end after sterilizing the part. Blood glucose levels were determined with a portable glucose meter. Insulin level was determined using a commercial ELISA kit (Mercodia, rat insulin).

### 2.13. Determination of Insulin Resistance

Insulin resistance was assessed by homeostasis model assessment (HOMA) [38] as a mathematical model describing the degree of insulin resistance, calculated from fasting plasma insulin and fasting plasma glucose concentration: (Fasting insulin × fasting glucose)/445.

### 2.14. Western Blot Analysis

To obtain information on the insulin sensitivity of the liver, small pieces of liver tissue (50 mg) were incubated in the presence of 1 μM insulin (Sigma-Aldrich) for 15 min at 37 °C.

All liver pieces were lysed in a buffer containing 150 mM NaCl, 50 mM Tris-HCl, 0.5% nonidet P-40, 0.5% sodium deoxycholate, 0.1% SDS, (pH 8), and Tissue Protease Inhibitor Cocktail (Sigma-Aldrich, 1:500, *v/v*). After 15 min of incubation, the lysates were centrifuged at 12,000× *g* for 30 min at 4 °C. The protein concentration in all lysates was evaluated by the biuret method. Immunoblotting was performed as previously reported [39] using the following commercially available antibodies: p-Akt (sc-377556, Santa Cruz, San Diego, CA, USA), Akt (sc-81434, Santa Cruz, San Diego, CA, USA), (for samples incubated and not with insulin), JNK (sc-7345, Santa Cruz, San Diego, CA, USA), p-JNK(sc-6254, Santa Cruz, San Diego, CA, USA); β-actin (A2066, Sigma-Aldrich, St. Louis, MO, USA). Secondary antibodies were purchased from Sigma- Aldrich (sc-2030, Santa Cruz, San Diego, CA, USA). The bands were visualized by the excellent chemiluminescent detection Kit (ElabScience, Microtech, Naples, Italy), according to the manufacturer’s instructions. Quantitative band densitometry was performed by analyzing ChemiDoc images or digital images of X-ray films exposed to immunostained membranes, and signal quantification was performed by Un-Scan-It gel software (Silk Scientific, Version 4.1, Oakmont Lane, Provo, UT, USA). To compare the protein expression levels, a standard control sample was run on each gel, and all values were compared with the control sample which was assigned the value 1.

### 2.15. RNA Isolation, RT-PCR and qPCR

Total RNA was isolated from the liver by tissue homogenization and TRIzol Reagent (Thermo Fisher Scientific, Waltham, MA, USA), according to the manufacturer’s instructions. The isolated total RNA was quantified with a NanoDrop spectrophotometer and was reverse transcribed using “High-Capacity cDNA Reverse Transcription kit” (Thermo Fisher Scientific, Waltham, Massachusetts, MN, USA, Cat# 4368813). Gene expression analysis was performed by quantitative PCR assays using BrightGreen qPCR MasterMix (Applied Biological Materials, Richmond, BC, Canada), according to the manufacturer’s instructions, on a CFX Connect Detection System (Bio-Rad, Hercules, CA, USA).

The specific primer pairs used for qPCR were designed using the Oligo 4.0 program and are listed in Table 1. The specificity of the amplification reactions was confirmed by melt curve analysis. Actb and B2m were selected as housekeeping genes and the arithmetic mean of their Ct was used as a reference for calculating ∆Ct. Relative expression analysis was performed using the 2^−ΔΔCt^ method. All qPCR reactions were performed in technical duplicate or triplicate.

### 2.16. Data Analysis

The data, expressed as means ± standard error, were analysed by the one -way ANOVA analysis of variance method, followed by Tukey’s pairwise comparison tests. Probability values (*p*) < 0.05 were considered significant. All analyses were performed using GraphPad Prism 8.0.2 (GraphPad Software, San Diego, CA, USA).

## 3. Results

### 3.1. Body Parameters

We first evaluated the effects of treatments on serum FT_3_ and FT_4_ levels, body weight ratio, and liver vitamin E levels. As expected, FT_3_ (Figure 1A) levels were significantly increased, and FT_4_ (Figure 1B) levels were significantly decreased in T_3_ treated animals regardless of vitamin E supplementation while were not affected by vitamin treatment. Heart weight/body weight ratio (Figure 1C) was increased by T_3_ treatment and significantly reduced by vitamin E supplementation in hyperthyroid animals. The effects of vitamin E on heart weight/body weight may depend on the fact that cardiac hypertrophy is partly due to oxidative stress [40].

The vitamin E content (Figure 1D) was increased by both thyroid hormone treatment and Vitamin E supplementation so that in the H + VE animals, the vitamin E content was the highest.

The increase in vitamin E levels in hyperthyroid animals could depend on the ability of the thyroid hormone to increase or mobilize endogenous reserves or induce greater assimilation of the vitamin from food [41].

### 3.2. Lipid and Protein Oxidative Damage and In Vitro Susceptibility to Oxidative Damage of Liver Homogenate and Isolated Mitochondria

To evaluate whether the treatments influenced liver oxidative stress we measured the levels of markers of oxidative damage to lipids (hydroperoxides, HP) and proteins (protein-bound carbonyls, CO) in homogenates (Figure 2A,B) and mitochondria (Figure 2C,D). Furthermore, we assessed the susceptibility to oxidative stress in vitro by determining the changes in hydroperoxide levels after an oxidative insult (ΔHP) in homogenates and mitochondria (Figure 2C,F). In both homogenates and mitochondria, HP, CO and ΔHP were significantly increased in hyperthyroid animals and significantly reduced by dietary supplementation of vitamin E, in both euthyroid and hyperthyroid rats. Our data indicate that vitamin E can reduce the levels of oxidative damage and the susceptibility to oxidative stress.

### 3.3. ROS Content, Mitochondrial ROS Release and NADPH Oxidase Activity

To provide information on the mechanisms underlying the increased level of oxidative damage and susceptibility to oxidative stress in hyperthyroid animals and the protective effects of vitamin E, we assessed total ROS content (Figure 3A), mitochondrial release of H_2_O_2_ during pyruvate plus malate-supported basal (state 4, Figure 3B) respiration and ADP-stimulated (state 3, Figure 3C) respiration, and NADPH oxidase activity (Figure 3D).

All measured parameters significantly increased in T_3_-treated rats. Vitamin E supplementation reduced total ROS content, mitochondrial ROS release and NOX activity regardless of thyroid status. However, in hyperthyroid animals all measured parameters remained higher than in euthyroid ones. These data suggest that the increase in oxidative damage in hyperthyroidism and the reduction seen after vitamin E supplementation may be related to the reduction in ROS production.

### 3.4. Antioxidant Enzymes Activity of Liver Homogenate

To have information on the antioxidant enzymes system’s contribution to the changes in oxidative damage levels, we measured the activities of the enzymes glutathione peroxidase (Figure 4A), glutathione reductase (Figure 4B), catalase (Figure 4C), and superoxide dismutase (Figure 4D). All the antioxidant enzymes activities increased in hyperthyroid animals. At the same time, vitamin E supplementation reduced the activities of GPX, GR and SOD in euthyroid rats and of GPX, catalase and SOD in hyperthyroid rats.

### 3.5. Oxygen Consumption of Liver Homogenate and Isolated Mitochondria

Information on the effects of treatments on the metabolic capacities of tissues and mitochondria was obtained by evaluating uncoupled and coupled respiration (state 4 and state 3, respectively) and the respiratory control ratio (RCR, an index of the coupling between electron transport chain flux and ATP synthesis, and of mitochondrial integrity) both in tissue homogenates (Figure 5A,B) and in isolated mitochondria (Figure 5D,E) (Figure 5). Overall, tissue and mitochondrial oxygen consumption with pyruvate plus malate as respiratory substrates were higher in T_3_ treated animals in both state 4 and state 3 respiration. Vitamin E reduced oxygen consumption in homogenate in state 4, and in both respiratory states in mitochondria in both euthyroid and hyperthyroid rats. However, oxygen consumption remained higher in H + VE than in H preparations. RCR was increased by vitamin E in euthyroid and hyperthyroid homogenates (Figure 5C) and hyperthyroid mitochondria (Figure 5F). Our results indicate that vitamin E can contain the increases in oxygen consumption induced by thyroid hormone in state 4.

### 3.6. Akt and JNK Activation

To highlight whether the increased levels of oxidative stress and susceptibility to oxidative stress were related to the development of insulin resistance, we evaluated the activation levels of Akt after treating liver tissue slices with insulin by determining the levels of P-Akt, Akt and the ratio P-Akt/Akt (Figure 6A). Furthermore, to understand the role played by JNK activation in the induction of insulin resistance, we evaluated its phosphorylation levels (Figure 6B).

The P-Akt/Akt ratio in non-insulin stimulated tissues did not show significant differences between groups. After insulin stimulation, P-Akt/Akt ratio was significantly reduced by T_3_ treatment and only partially restored by vitamin E supplementation remaining lower than in the euthyroid controls. These results suggest that oxidative damage reduced Akt activation.

The decreases in hepatic insulin sensitivity induced by T_3_ treatment were accompanied by a significant increase in the tissue content of phosphorylated JNK, which was attenuated by vitamin E supplementation.

These results suggest that oxidative stress induced by hyperthyroidism may be responsible for increased insulin resistance.

### 3.7. Glycaemic Homeostasis Parameters

To understand whether changes in hepatic insulin resistance were accompanied by changes in systemic insulin sensitivity, we evaluated fasting glycemia (Figure 7A) and insulinemia (Figure 7B), and the HOMA index (Figure 7C). All measured parameters were not significantly affected by vitamin E supplementation in euthyroid rats while were significantly increased by T_3_ treatment. Such changes were counteracted by vitamin E supplementation so that glycemia was restored to euthyroid levels.

### 3.8. Gene Expression Analysis of Slc2a1, Slc2a2, Pparg, Irs2, Ppara, Cd36 and Il1b

To identify putative genes affected in hyperthyroid rats and potentially targeted by vitamin E, we analysed the expression of genes encoding glucose transporters (e.g., *Slc2a1* and *Slc2a2*, Figure 8A,B), genes involved in hepatic glucose/lipid homeostasis and insulin signalling (e.g., *Ppara*, *Pparg*, *Cd36*, *Irs2* (Figure 8C–F), as well as *Il1b* (Figure 8G) with a known inflammatory role related to insulin response. As shown in Figure 8, T_3_-treated rats displayed a significant upregulation of *Slc2a2*, *Cd36* and *Il1b*. Conversely, *Pparg* was markedly, and *Slc2a1* was slightly reduced by T_3_ treatment. Vitamin E supplementation increased *Pparg* expression in euthyroid animals, also weakening the decrease induced by T_3_-treatment in hyperthyroid animals. Moreover, vitamin E enhanced the increase of *Slc2a2* while attenuating the increase in of *Cd36* in hyperthyroid animals.

## 4. Discussion

In previous works we found that ten days of T_3_ administration are enough to induce thyroid hormone’s induced alterations, such as increased heart rate which is associated to a decreased of the ventricular action potential duration [25]. Both vitamin E and *N*-acetyl cysteine attenuate the thyroid hormone -induced reduction of ventricular action potential duration [42] and hear rate increase [25], suggesting a ROS role in such hyperthyroidism-linked alterations.

In the present work, we have defined a direct relationship between hepatic insulin resistance and oxidative stress in the hyperthyroid state. Treatment for ten days with thyroid hormone induces an increase in rat liver content of markers of oxidative damage to lipids and proteins. The increase in oxidative damage mainly depends on the increase in ROS content to which both an increased activity of hepatic NADPH oxidase, and an increase in the release of hydrogen peroxide during mitochondrial respiration contribute. On the other hand, treatment with thyroid hormone also induces biochemical changes in macromolecules, and, in particular, in phospholipids, in which the peroxidability index increases [43] which explains, at least in part, increased levels of lipid oxidation markers.

As expected, dietary supplementation with vitamin E reduces oxidative damage in liver preparations. The main mechanism underlying the protective effect of vitamin E against oxidative damage depends on the direct scavenging action on lipid and protein-derived peroxyl radicals in highly organized membrane structures. However, our data indicate that vitamin E can exert its protective effect by reducing the ROS content and both NOX activity and the release of mitochondrial ROS. The increase in NOX activity in the liver of T_3_-treated animals agrees with previous data [44] and the slight but significant reduction in NOX activity in rats fed vitamin E may depend on the ability of the antioxidant to reduce the translocation of the cytosolic component of the enzyme, p47^phox^, to the membrane, as suggested by in vitro studies [45,46]. Vitamin E-induced reduction in mitochondrial H_2_O_2_ release rate may depend on the ability of its benzene ring to regulate the mitochondrial generation of superoxide and hydrogen peroxide. Vitamin E can modulate mitochondrial production and superoxide levels by preventing electron leakage, directly regulating superoxide generation systems and/or eliminating superoxide generated [47]. In this way, vitamin E reduces the possibility that the superoxide formed when electrons escape the mitochondrial electron transport chain, can trigger a series of peroxidation chain reactions [41].

Interestingly, the changes in mitochondrial oxidative stress linked to differences in hepatic ROS content appear to be responsible for the increase in basal oxygen consumption (state 4 of respiration). Indeed, T_3_ treatment increases whereas vitamin E administration reduces such consumption in the euthyroid and hyperthyroid animals. In contrast, vitamin E does not affect the increase in ADP-stimulated oxygen consumption in the hyperthyroid liver, which suggests that it does not interfere with the hormone-induced increase in metabolic capacity.

It is worth noting that the increase in oxidative stress also depends on the increased susceptibility to oxidants, which may, in turn, depend either on the reduced efficiency of the antioxidant system and/or on the increased susceptibility to oxidants of the biomolecules. Both conditions occur in hyperthyroid animals. Indeed, although our data indicate that the activities of the main antioxidant enzymes increase in hyperthyroid animals, the effectiveness of the antioxidant system decreases as demonstrated by the decrease in the total antioxidant capacity previously reported [11] and as suggested by the increased susceptibility to oxidants reported here. The increased activity of antioxidant enzymes in hyperthyroid rats appears to be a compensatory response to increased oxidative stress, as suggested by the increased expression of antioxidant enzymes in an in vitro system after H_2_O_2_ treatment [47,48,49,50,51]. It has been suggested that the hyperthyroid state induces the transcription of antioxidant enzymes through ROS-induced activation of the nuclear factor erythroid 2-related factor [48]. Our results show reduced activation of GPX, catalase and SOD in rats supplemented with vitamin E, and support the role of ROS in the activation of antioxidant enzymes.

Although the mechanisms by which high H_2_O_2_ production impairs insulin signaling are not fully characterized, the link between ROS production and IR has been ascribed to alterations in various intracellular signaling pathways. As it is known, Insulin binding to the insulin receptor recruits phosphoinositide-3-phosphate kinase through insulin receptor substrate (IRS) and generates phosphatidylinositol (3,4,5)-trisphosphate (PIP3). PIP3 activates the protein kinase 1 which, in turn, initiates the activation of Akt through phosphorylation on Thr^308^ [49]. For the full activation of Akt it is necessary additional phosphorylation due to the mechanistic target of rapamycin complex 2 (mTORC2) at Ser^473^ [50]. Akt has several substrates and, in the liver, the activation of Akt opens several avenues for the control of glucose and lipid homeostasis.

Activated Akt inhibits both glycogenolysis and gluconeogenesis through multiple downstream pathways including glycogen synthase kinase 3 and forkhead box protein O1 (FoxO1) [51]. Furthermore, G6P activates the carbohydrate response-element binding protein (ChREBP), which activates lipogenesis together with SREBP1c. Akt inhibits FoxO1, resulting in an inhibition of gluconeogenesis by suppressing the expression of the proteins glucose-6-phosphatase (G6pc) and phosphoenolpyruvate carboxykinase (Pck1) [51].

We found that Akt activation is significantly reduced in a condition of oxidative stress induced by T_3_ treatment and partially restored by antioxidant supplementation. Changes in hepatic insulin sensitivity parallel changes in fasting glucose and insulin levels and the HOMA index. Hence, our data are consistent with the idea that food supplementation with vitamin E can protect against the alteration of insulin sensitivity induced by ROS.

To obtain information on the possible actors involved in the effects of vitamin E on insulin sensitivity, we measured the activation levels of JNK, one of the most studied factors in the obesity models of IR [52]. Both inhibition and total or partial deletion of JNK attenuate the onset of IR. For example, in a rat model of non-alcoholic fatty liver disease (NAFLD) induced by a high-fat diet (HFD), treatment with a JNK inhibitor attenuated insulin resistance [53]. Alteration of glycaemic homeostasis related to P-JNK levels was also reported in obese mice, in which blood glucose and insulinemia were significantly improved in JNK^+/−^ mice, but even more so in JNK^−/−^ mice [54].

JNK is considered the primary signal transducer that regulates the physiological response to various cellular stressors, including ultraviolet (UV) rays, genotoxic damage, oxidative stress, endoplasmic reticulum stress, long-chain saturated fatty acids, inflammatory cytokines, and microbial by-products. The stressors activate JNK, a protein kinase that mediates phosphorylation and activation of the cJun transcription factor [52].

We found that in hyperthyroid rats, kinase phosphorylation is significantly increased compared to control animals and is attenuated by antioxidant supplementation. Thus, our results agree with the observation that endogenous ROS act as potent inducers of JNK [55], confirming the correlation between increased oxidative stress and JNK phosphorylation.

In line with the reduced insulin response assessed in T_3_-treated rats, we revealed reduced expression of the *Pparg* gene. Indeed, PPARγ crucially regulates hepatic lipid uptake and liver-specific *Pparg*-null mice exhibit aggravated hyperlipidaemia and insulin resistance [56,57]. Interestingly, vitamin E can induce *Pparg* expression in both control and hyperthyroid rats, counteracting the downregulation induced by T_3_ treatment and restoring its expression to physiological levels. Therefore, the ability of vitamin E supplementation to reduce glycaemic and insulinemic peaks in hyperthyroid rats could underlie, at least in part, the effects on *Pparg* expression. Furthermore, a marked induction of the expression of another member of the PPAR family that regulates hepatic lipid metabolism, namely *Ppara*, was exerted by vitamin E in the T_3_-treated rats, while a milder effect is observed in the control rats. Since anti-fibrotic and anti-inflammatory effects for PPARα in the liver have also been found [57,58,59], the ability of vitamin E to induce *Ppara* in the hyperthyroid group could contribute to the observed improvement of insulin response and redox state. Furthermore, we also revealed marked upregulation of *Cd36* following the induction of hyperthyroidism. Conflicting results have been reported on *Cd36* expression in insulin resistant population and animal models, indicating a complex impact of CD36 deficiency on insulin homeostasis [60,61,62,63,64]. However, our data are consistent with the association between Cd36 upregulation in the liver and insulin resistance, hyperinsulinemia, and steatosis [62], as well as with the induced increase in CD36 levels from inflammatory stress [65]. Furthermore, vitamin E can greatly counteract this induction, although *Cd36* levels remain upregulated compared to controls. In particular, this effect is in line with the ability of vitamin E to significantly reduce, but not completely rescue, the alteration of the insulin response and redox state. Furthermore, the induction of *Slc2a2* expression induced by vitamin E supplementation in hyperthyroid rats is also consistent with the improvement in basal glycaemic levels. Finally, the expression analysis also indicated that the alterations in glucose homeostasis and insulin response evaluated in hyperthyroid rats, as well as the beneficial effects of vitamin E, are not related to any expression change of *Slc2a1* and *Irs2*. Otherwise, the variation of the genes *Pparg*, *Ppara*, *Cd36* and *Slc2a2* could be the basis of the observed metabolic phenotypes.

## 5. Conclusions

Our data suggest that in conditions of oxidative stress such as that found in a hyperthyroid state, antioxidant supplementation can attenuate the alteration of insulin homeostasis by reducing the release of ROS. Furthermore, the observation that the effects of vitamin E on insulin sensitivity are mediated by a reduced ROS content also due to the reduced release of mitochondrial ROS without interfering with the increase in metabolic capacity induced by T_3_, agrees with the observation that insulin resistance depends mainly on ROS release in itself, and not on the dysfunction of the mitochondrial electron transport chain [66].

However, even if our data suggest a beneficial effect of vitamin E supplementation in animals treated with T_3_ on alteration of glycaemic homeostasis induced by oxidative stress, with low effects on control animals, it is necessary to emphasize that indiscriminate vitamin E supplementation can induce detrimental effects on the various physiological processes, including glycaemic homeostasis. Indeed, chronic supplementation with high doses of antioxidants for control animals has been shown to hinder insulin signalling in liver tissues, and this has been associated with impaired glucose tolerance, insulin resistance and hyperglycaemia [67]. We can conclude that hyperthyroidism-induced oxidative stress is involved in vivo in the onset of insulin resistance, but other studies are needed to understand under what circumstances the strengthening of the antioxidant defences can be a weapon of prevention.

Moreover, we must underline that we used as experimental model male rats and since it has been reported that thyroid dysfunction is more common in female than in male humans [68] we cannot exclude that insulin resistance in hyperthyroid female can respond in a different way to antioxidant supplementation. Further studies employing female rats will help to clarify this aspect.

## Figures and Tables

**Figure 1 antioxidants-11-01295-f001:**
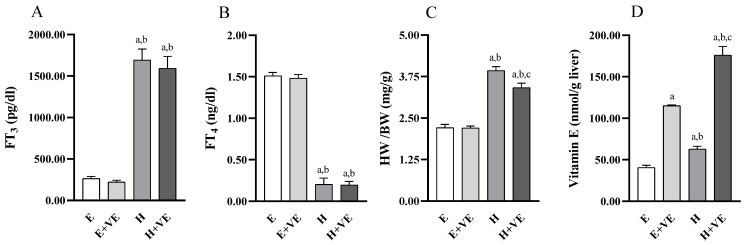
Body parameters. FT_3_ plasma levels (**A**), FT_4_ plasma levels (**B**), heart weight/body weight ratio (**C**), and Vitamin E content in liver tissue (**D**). Values are means ± SEM of eight determinations from different rats. Each determination is the mean of three independent measures for the data reported in panels A, B and D. E, euthyroid; E + VE, euthyroid fed vitamin E supplemented diet; H, hyperthyroid; H + VE, hyperthyroid fed vitamin E supplemented diet. ^a^ significant vs. E. ^b^ significant vs. E + VE. ^c^ significant vs. H. The level of significance was chosen as *p* < 0.05.

**Figure 2 antioxidants-11-01295-f002:**
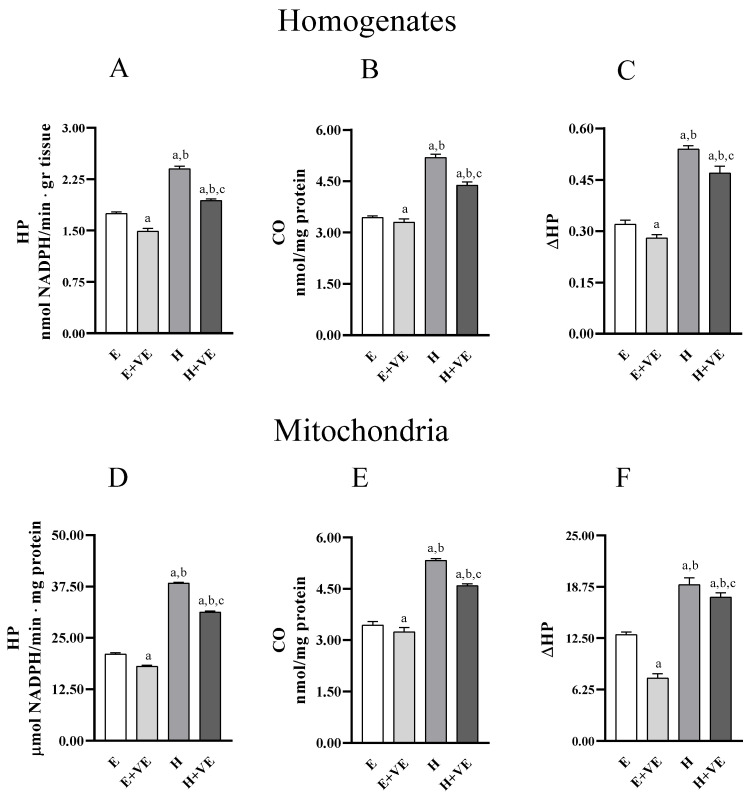
Lipid (hydroperoxides, HP, panels (**A**,**D**)) and protein (protein bound carbonyls, CO, panels (**B**,**E**)) oxidative damage markers; and in vitro response to oxidative stress (ΔHP, panels (**C**,**F**)), in liver homogenate (upper panels) and isolated mitochondria (lower panels) from E, E + VE, H and H + VE rats. Values are means ± SEM of eight determinations from different rats. Each determination is the mean of three independent measures. E, euthyroid; E + VE, euthyroid fed vitamin E supplemented diet; H, hyperthyroid; H + VE, hyperthyroid fed vitamin E supplemented diet. ^a^ significant vs. E. ^b^ significant vs. E + VE. ^c^ significant vs. H. The level of significance was chosen as *p* < 0.05.

**Figure 3 antioxidants-11-01295-f003:**
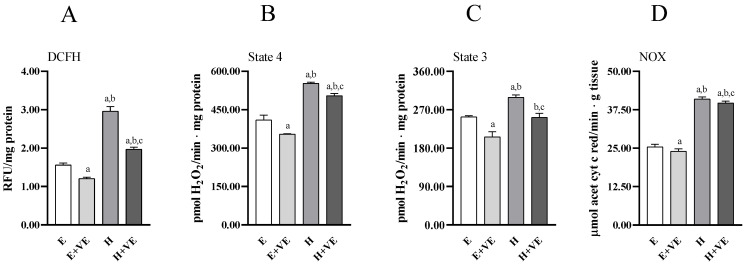
Total ROS content (**A**), mitochondrial H_2_O_2_ release during state 4 (**B**) and state 3 (**C**) of respiration, and NADPH oxidase activity (**D**) in liver from E, E + VE, H and H + VE rats. Values are means ± SEM of eight determinations from different rats. Each determination is the mean of three independent measures. E, euthyroid; E + VE, euthyroid fed vitamin E supplemented diet; H, hyperthyroid; H + VE, hyperthyroid fed vitamin E supplemented diet. ^a^ significant vs. E. ^b^ significant vs. E + VE. ^c^ significant vs. H. The level of significance was chosen as *p* < 0.05.

**Figure 4 antioxidants-11-01295-f004:**
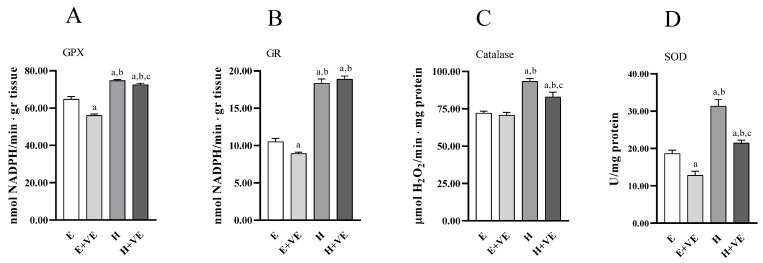
Antioxidant enzyme activities. GPX (**A**), GR (**B**) catalase (**C**), and SOD activity (**D**) in liver from E, E + VE, H and H + VE rats. Values are means ± SEM of eight determinations from different rats. Each determination is the mean of three independent measures. E, euthyroid; E + VE, euthyroid fed vitamin E supplemented diet; H, hyperthyroid; H + VE, hyperthyroid fed vitamin E supplemented diet. ^a^ significant vs. E. ^b^ significant vs. E + VE. ^c^ significant vs. H. The level of significance was chosen as *p* < 0.05.

**Figure 5 antioxidants-11-01295-f005:**
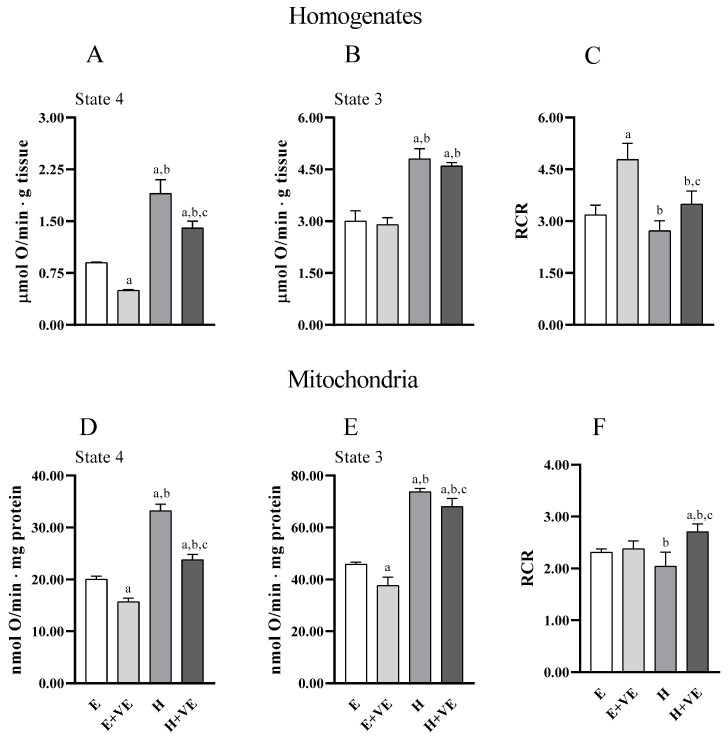
State 4 and State 3 oxygen consumption, and respiratory control ratio (RCR) detected in pyruvate plus malate supplemented homogenate (**A**–**C**) and mitochondria (**D**–**F**). Values are means ± SEM of eight determinations from different mice. Each determination is the mean of three independent measures. E, euthyroid; E + VE, euthyroid fed vitamin E supplemented diet; H, hyperthyroid; H + VE, hyperthyroid fed vitamin E supplemented diet. ^a^ significant vs. E. ^b^ significant vs. E + VE. ^c^ significant vs. H. The level of significance was chosen as *p* < 0.05.

**Figure 6 antioxidants-11-01295-f006:**
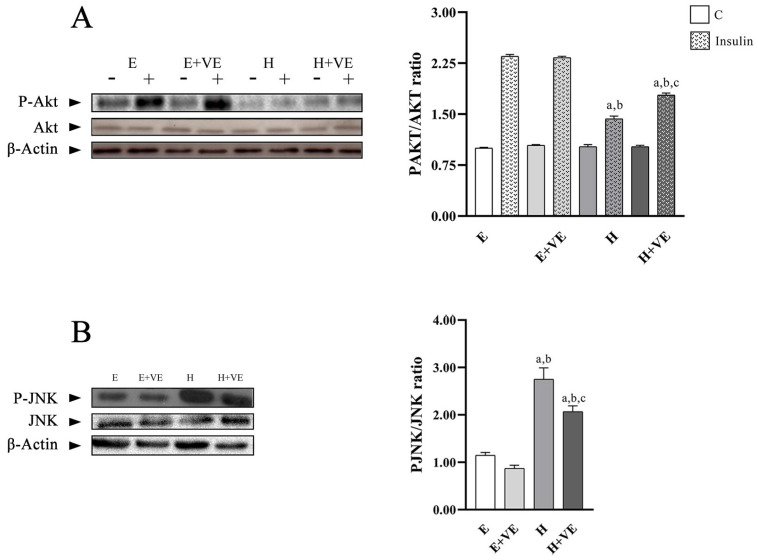
(**A**) left, representative western blot images of P-Akt, Akt and β-Actin in the absence (−) and in the presence (+) of insulin in rat liver, each line contained 30 μg of proteins; right, P-Akt/Akt ratio in the absence (C) and in the presence of insulin (insulin). (**B**) left, representative western blot images of P-JNK, JNK and β-Actin in rat liver, each line contained 30 μg of proteins; right, quantification of data. Values are means ± SEM of three determinations from different rats. Each determination is the mean of three independent measures. E, euthyroid; E + VE, euthyroid fed vitamin E supplemented diet; H, hyperthyroid; H + VE, hyperthyroid fed vitamin E supplemented diet. ^a^ significant vs. E. ^b^ significant vs. E + VE. ^c^ significant vs. H. The level of significance was chosen as *p* < 0.05.

**Figure 7 antioxidants-11-01295-f007:**
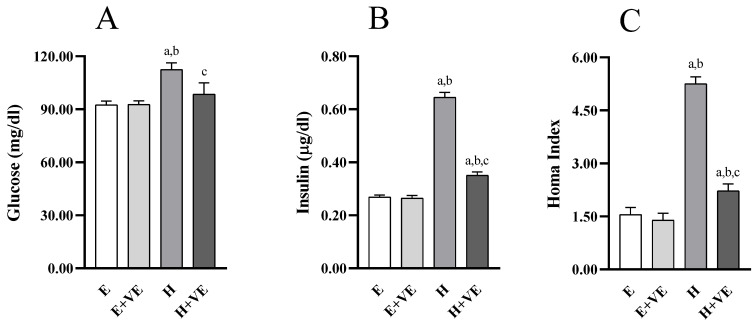
Parameters of glucose homeostasis: Blood glucose fasting levels (**A**), plasma insulin fasting levels (**B**), HOMA Index (**C**). Values are means ±SEM of eight determinations from different rats Each determination is the mean of three independent measures. E, euthyroid; E + VE, euthyroid fed vitamin E supplemented diet; H, hyperthyroid; H + VE, hyperthyroid fed vitamin E supplemented diet. ^a^ significant vs. E. ^b^ significant vs. E + VE. ^c^ significant vs. H. The level of significance was chosen as *p* < 0.05.

**Figure 8 antioxidants-11-01295-f008:**
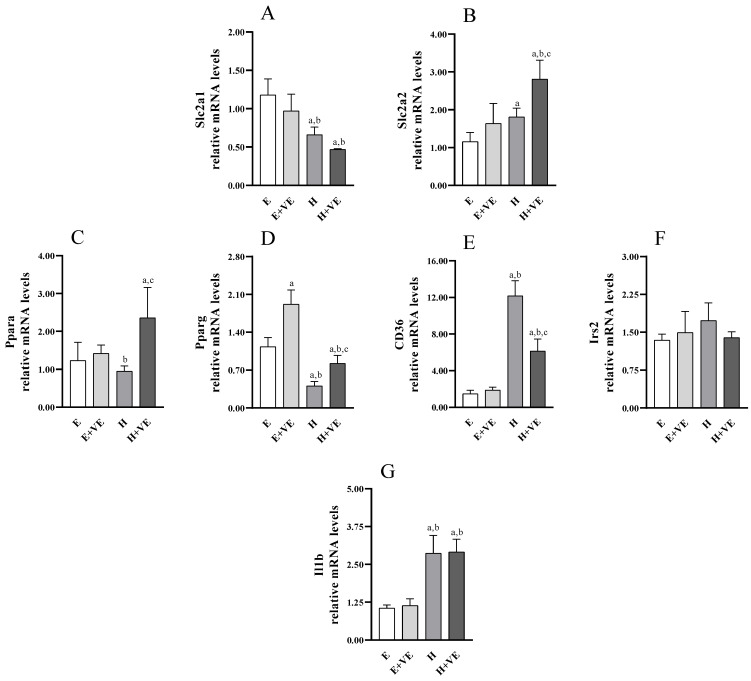
Relative mRNA quantification by qPCR of indicated genes ((**A**) *Slc2a1*; (**B**) *Slc2a2*; (**C**) *Ppara*; (**D**) *Pparg*; (**E**) *Cd36*; (**F**) *Irs2*; (**G**) *Il1b*). Values are means ±SEM of three determinations from different rats. Each determination is the mean of three independent measures. E, euthyroid; E + VE, euthyroid fed vitamin E supplemented diet; H, hyperthyroid; H + VE, hyperthyroid fed vitamin E supplemented diet. ^a^ significant vs. E. ^b^ significant vs. E + VE. ^c^ significant vs. H. The level of significance was chosen as *p* < 0.05.

**Table 1 antioxidants-11-01295-t001:** Sequences of oligonucleotides used as primers in qPCR reactions.

Gene Symbol	c	Sense Primer (5′---3′)	Antisense Primer (5′---3′)
*Cd36*	NM_031561.2	TTACTGGAGCCGTTATTGGTG	CCTTGATCTTGCTGCTATTCT
*Il1b*	NM_031512.2	AGGCTGACAGACCCCAAAAG	AAGCTCCACGGGCAAGACAT
*Irs2*	NM_001168633.1	GCCAGCACCTACGCAAGCA	AGCCCTGCCTCTTGGTTCC
*Ppara*	NM_013196.2	CCACTTGAAGCAGATGACCT	CATTGCCAGGGGACTCATCT
*Pparg*	NM_013124.3; NM_001145366.1; NM_001145367.1	GTCGGATCCACAAAAAGAGTA	TTTGTCTGTTGTCTTTCCTGT
*Slc2a1*	NM_138827.2	GCGGGCTGTGCTGTGCTC	CCACAGCAACAGCAGCAG
*Slc2a2*	NM_012879.2	GGGAAGAAGAGACTGAAGGA	CTTCCAGCAATGATGAGAGC

## Data Availability

All of the data is contained within the article.

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
