# Peer review of "Hepatic Insulin Resistance in Hyperthyroid Rat Liver: Vitamin E Supplementation Highlights a Possible Role of ROS"

_antioxidants, 2022, doi:10.3390/antiox11071295_

Round 1

Reviewer 1 Report

The manuscript by Fasciolo G et al investigated the effect of antioxidant vitamin E on oxidative damage and insulin resistance in the liver of experimental hyperthyroid rats. The main findings are 1) Hyperthyroid increased oxidative damage markers, ROS contents and release, oxygen consumption and JNK activation in the liver, which were restored (partially) by Vitamin E treatment. 2) Hyperthyroid affected the expression of genes encoding glucose transporters e.g. Slc2a2 and involved in hepatic glucose homeostasis e.g. Pparg that were further modulated by Vitamin E. The authors conclude that appropriate dose of Vitamin E supplementation could improve insulin sensitivity in the liver of hyperthyroid animals. This study is very interesting and presented many relevant data to support their conclusion. However, some key points are need to be addressed and clarified. 

Major points: 

1.     Title. The title is rather vague and it could be more specific and indicates an animal study and vitamin E.

2.     Abstract. The results in the abstract were only briefly described. 

3.     Line 90, the study was conducted only in male rats while hyperthyroidism is more common in females than males. Some justifications are required. The number of animals used in each group should be clearly stated. 

4.     Line 92 and 96, how was the dose of Vitamin E determined?

5.     Line 248-250, what was a standard control sample?

6.     In all figure legend, please the number of technical replicates and biological replicates (e.g. tissues or animals).

7.     Please show all data points in all bar graphs.

8.     Line 336-337, Vitamin E decreased the activity of antioxidant enzymes GPX, catalase and SOD. Is it contradictory to the effect of Vitamin E reducing oxidative stress?

9.     Figure 6A, the loading control beta-action are uneven for different groups and the Akt band for H+ group did not seem to develop well and the authors may consider choosing another representative image. The amount of proteins loaded in each lane should be indicated in the figure legend. 

10.  As shown in the Figure 7B, Vitamin E restored the insulin level in hyperthyroid rats. Does it indicate that Vitamin E modulate the release of insulin from pancreatic beta cells?

11.  Line 410, was supposed Pparg rather than Cd36 significantly reduced by T3 treatment as shown in the figure 8D?

12.  Line 518-520, the expression of Ppara was not altered between H and H+VE group as shown in the figure 8C, which was inconsistent with the description here.

13. Akt WB image in the original blot image did not seem to come from the same blot.

Minor points: 

1.     Please check some typos or grammar errors, e.g., line 17-18 “it conceivable role of ROS”; line 41, “able to impair”; line 476, “For the full activation of Akt is necessary additional”, line 547, “di per se”. 

2.     Line 34, insulin agonistic and antagonistic actions

3.     Line 262, geometric mean of two reference gene?

4.     The labeling of figures “A, B, C…” was not indicated in the text. 

5.     Figure 1C, Y-axis label should indicate “Vitamin E concentration”.

Author Response

The manuscript by Fasciolo G et al investigated the effect of antioxidant vitamin E on oxidative

damage and insulin resistance in the liver of experimental hyperthyroid rats. The main findings are

1) Hyperthyroid increased oxidative damage markers, ROS contents and release, oxygen

consumption and JNK activation in the liver, which were restored (partially) by Vitamin E treatment.

2) Hyperthyroid affected the expression of genes encoding glucose transporters e.g. Slc2a2 and

involved in hepatic glucose homeostasis e.g. Pparg that were further modulated by Vitamin E. The

authors conclude that appropriate dose of Vitamin E supplementation could improve insulin

sensitivity in the liver of hyperthyroid animals. This study is very interesting and presented many

relevant data to support their conclusion. However, some key points are need to be addressed and

clarified.

Many thanks to the reviewer for the careful review that allowed us to correct some errors that we had missed.

Major points:

  1. Title. The title is rather vague and it could be more specific and indicates an animal study and

vitamin E.

the title has been changed in “Hepatic insulin resistance in hyperthyroid rat liver: vitamin E supplementation highlights a possible role of ROS”

  1. The results in the abstract were only briefly described.

The journal “Antioxidants” requires an that abstract does not exceed 200 words. Therefore, the results cannot be extensively described.  However, in the revised version of our manuscript a tentative has ben done to give more information

  1. Line 90, the study was conducted only in male rats while hyperthyroidism is more common in

females than males. Some justifications are required. The number of animals used in each group

should be clearly stated.

We used rats of the same gender as an animal model of hyperthyroidism because we have long known that male and female rats have different responses to different pathophysiological conditions.

Venditti P, Piro MC, Artiaco G, Di Meo S. Effect of exercise on tissue antioxidant capacity and heart electrical properties in male and female rats. Eur J Appl Physiol Occup Physiol. 1996;74(4):322-9. doi: 10.1007/BF02226928. PMID: 8911824.

Di Meo S, de Martino Rosaroll P, Venditti P, Balestrieri M, De Leo T. Action potential configuration in heart papillary muscles from female rats in different thyroid states. Arch Physiol Biochem. 1997 Feb;105(1):58-65. doi: 10.1076/apab.105.1.58.13146. PMID: 9224547.

The aim of our study  was to understand whether ROS play a role in insulin resistance  induced by hyperthyroidism and not to investigate the heterogeneous response between genders that here we consciously avoided using only rats of the same gender.

  1. Line 92 and 96, how was the dose of Vitamin E determined?

In Materials and Methods, we wrote:

“It has been found that this supplementation dose can reduce oxidative damage in sedentary and trained rats [24] and attenuate the changes induced by thyroid hormone on cardiac electrical activity in rats [25].”

So, we used a supplementation dose that showed to be able to increase the tissue levels of α- tocopherol and showed to be efficient in the above-mentioned conditions.

In the present version of the manuscript, we write “this supplementation dose” to avoid confusion.

  1. Line 248-250, what was a standard control sample?

We always load on the gels the same euthyroid control, in a lateral well, and we attribute the value 1 to the ratio between the intensity of the actin band of this control and the band of the protein of interest, to compare all the other ratios actin/interest band with this last.

  1. In all figure legend, please the number of technical replicates and biological replicates (e.g. tissues

or animals).

The number of biological replicates was already present in our work. We have added the number of technical replicates.

  1. Please show all data points in all bar graphs.

Due to the way we built our graphs, adding all the data points would mean doing it all over again. We built our bar charts with the mean and standard error of the mean. I do not think it is essential unless the reviewer wants to verify the veracity of the data shown in the figures.

  1. Line 336-337, Vitamin E decreased the activity of antioxidant enzymes GPX, catalase and SOD. Is it

contradictory to the effect of Vitamin E reducing oxidative stress?

We discussed this aspect in the section discussion:

“The increased activity of antioxidant enzymes in hyperthyroid rats appears to be a compensatory response to increased oxidative stress, as suggested by the increased expression of antioxidant enzymes in an in vitro system after H2O2 treatment [46-50].  It has been suggested that the hyperthyroid state induces the transcription of antioxidant enzymes through ROS- induced activation of the nuclear factor erythroid 2-related factor [47]. Our results show reduced activation of GPX, catalase and SOD in rats supplemented with vitamin E, and support the role of ROS in the activation of antioxidant enzymes. “

  1. Figure 6A, the loading control beta-action are uneven for different groups and the Akt band for H+

group did not seem to develop well and the authors may consider choosing another representative

image. The amount of proteins loaded in each lane should be indicated in the figure legend.

In the present version of the manuscript, we used images for beta actin and Akt obtained after a longer period of exposure to improve the clarity of the bands.

Moreover, we added the amount of proteins loaded in each lane in the figure legends.

  1. As shown in the Figure 7B, Vitamin E restored the insulin level in hyperthyroid rats. Does it indicate

that Vitamin E modulate the release of insulin from pancreatic beta cells?

Vitamin E administration to hyperthyroid animals did not restore insulin level to the euthyroid ones but reducing glucose plasma level reduces insulin release. Therefore, the reduction of insulin levels mainly depends on the reduced level of glucose in H+VE rats.

  1. Line 410, was supposed Pparg rather than Cd36 significantly reduced by T3 treatment as shown in

the figure 8D?

Many thanks to the reviewer for looking closely at Figure 8. We mistakenly entered the wrong histogram for Ppara. In the revised version of the work, we changed the figure

  1. Line 518-520, the expression of Ppara was not altered between H and H+VE group as shown in the

figure 8C, which was inconsistent with the description here.

In the present version of the manuscript, we have inserted the correct histogram of Ppara. Thanks to the reviewer for his attention    

  1. Akt WB image in the original blot image did not seem to come from the same blot.

The Akt image in Figure 6A and in the blot was the same image but obtained at different development times. In the revised version of the manuscript, we used the image obtained with a longer development time.

Minor points:

1 Please check some typos or grammar errors, e.g., line 17-18 “it conceivable role of ROS”; line 41,

“able to impair”; line 476, “For the full activation of Akt is necessary additional”, line 547, “di per se”.

We checked and corrected

2 Line 34, insulin agonistic and antagonistic actions

   We checked and corrected

3 Line 262, geometric mean of two reference gene?

Using “average” we referred to “arithmetic mean”. We thank the reviewer for the question, which allowed us to better clarify the sentence, specifying that we calculated arithmetic mean.

4 The labeling of figures “A, B, C…” was not indicated in the text.

We indicated in the test the panel labelling

5 Figure 1C, Y-axis label should indicate “Vitamin E concentration”.

We indicate in the figure 1 “Vitamin E” on the Y-axis  

Reviewer 2 Report

The review by Gianluca Fasciolo et al is devoted to the well known medical problem of insulin resistance induced by hyperthyroidism. The relationship between the development of insulin resistance and oxidative stress in the liver tissue of hyperthyroid rats is analyzed in all details including the level of ROS production, ROS-induced damage of lipids and proteins, the activities of antioxidant enzymes, the ability of the tissue to resist the oxidative insult and impair of the insulin associated signaling pathways. The authors have convincingly shown that if, simultaneously with injections of T3, rats are fed for 10 days with food enriched with vitamin E, oxidative damage and its dangerous consequences can be significantly reduced. The methods used in the work are adequate and described in details. At the same time I did not find the number of animals in each group analyzed by the authors. This flaw should be fixed. Minor technical disadvantages include the designation of the control group with the letter E, which is associated with vitamin E. Possibly it should be better C-group (control) in future. Also for designation dL (Fig.1) it would be more correct to write dl. With the exception of an unknown number of animals in groups, the remaining shortcomings are small and do not detract from the overall good impression of the work. 

Author Response

  • At the same time, I did not find the number of animals in each group analyzed by the authors. This flaw should be fixed.

I indicated in the legend of the figures the number of rats used for each group. However, in the revised version of the paper I wrote in the paragraph 2.1 : “From day 80 to day 90, half of the 32 animals recruited for the study received, once a day i.p. injection of triiodothyronine (T3) (Sigma-Aldrich) (50 μg / 100 body weight) [23].”

  • Minor technical disadvantages include the designation of the control group with the letter E, which is associated with vitamin E. Possibly it should be better C-group (control) in future.

I understand that using E can confuse the reader, but for me E means Euthyroid, and VE vitamin E. I would like to leave the notation unchanged in this work. I will use other letters in the future

  • Also for designation dL (Fig.1) it would be more correct to write dl.

Many thanks for having detected the error it had completely escaped me. I corrected all the figures with dL

With the exception of an unknown number of animals in groups, the remaining shortcomings are small and do not detract from the overall good impression of the work. 

Reviewer 3 Report

This original article entitled “Hepatic insulin resistance in hyperthyroidism: the possible role of ROS” provided evidences to reveal that hyperthyroidism increasing ROS results in tissue oxidative stress, that would be considered a hallmark for leading to IR. It is an interesting study but some criticism remained to be addressed.

Major comments,

1.     Experiments including animal data should be provided the ethical approval by the authors’ host organization; the approval number should also be provided.

2.     Line 93-94, … animal was received i.p. injections of triiodothyronine (T3) (50 μg / 100 body weight). Is the dosage 50 μg / 100 g body weight? Is it single dose or multiple doses? What is the frequency?

3.     In Figure 1A, to show the activity of administrated T3 in individual, T4 level should also be detected.

4.     In this study, the induction of T3 for 10 days is a short-term experiment. The temporary physiological changes should be recovered while T3 withdraws. However, hyperthyroidism is a long-term metabolic alternation including multiple pathological processes, whether the mechanisms are similar with the current results needed to be discussed more.

Minor comments,

1.     Some typo errors still need to proofread.

Author Response

  1. Experiments including animal data should be provided the ethical approval by the authors’ host organization; the approval number should also be provided.

Thanks to the reviewer for the suggestion. We added the protocol number of the authorization received from the Italian Ministry of Health (836/2019 PR)

  1. Line 93-94, … animal was received i.p. injections of triiodothyronine (T3) (50 μg / 100 body weight). Is the dosage 50 μg / 100 g body weight? Is it single dose or multiple doses? What is the frequency?

We treated animals with daily doses of T3. We have changed as suggested by the reviewer to better define our protocol

  1. In Figure 1A, to show the activity of administrated T3 in individual, T4 level should also be detected.

In the present version of the manuscript, we have added the plasma FT4 levels.

  1. In this study, the induction of T3 for 10 days is a short-term experiment. The temporary physiological changes should be recovered while T3 withdraws. However, hyperthyroidism is a long-term metabolic alternation including multiple pathological processes, whether the mechanisms are similar with the current results needed to be discussed more.

Thanks to the reviewer for the comment.  We added some information on the effects of antioxidant treatment on other hyperthyroidism-induced alterations, such as heart rate and ventricular action potential duration.

Minor comments,

  1. Some typo errors still need to proofread.

Typo errors have been corrected

Round 2

Reviewer 1 Report

I would like to thank the authors for responding to my initial comments, but I still have some concerns below.

  1. Line 90, the study was conducted only in male rats while hyperthyroidism is more common in

females than males. Some justifications are required. The number of animals used in each group

should be clearly stated.

We used rats of the same gender as an animal model of hyperthyroidism because we have long known that male and female rats have different responses to different pathophysiological conditions.

Venditti P, Piro MC, Artiaco G, Di Meo S. Effect of exercise on tissue antioxidant capacity and heart electrical properties in male and female rats. Eur J Appl Physiol Occup Physiol. 1996;74(4):322-9. doi: 10.1007/BF02226928. PMID: 8911824.

Di Meo S, de Martino Rosaroll P, Venditti P, Balestrieri M, De Leo T. Action potential configuration in heart papillary muscles from female rats in different thyroid states. Arch Physiol Biochem. 1997 Feb;105(1):58-65. doi: 10.1076/apab.105.1.58.13146. PMID: 9224547.

The aim of our study  was to understand whether ROS play a role in insulin resistance  induced by hyperthyroidism and not to investigate the heterogeneous response between genders that here we consciously avoided using only rats of the same gender.

The authors could include this as a limitation of the study in the discussion.

  1. Figure 6A, the loading control beta-action are uneven for different groups and the Akt band for H+

group did not seem to develop well and the authors may consider choosing another representative

image. The amount of proteins loaded in each lane should be indicated in the figure legend.

In the present version of the manuscript, we used images for beta actin and Akt obtained after a longer period of exposure to improve the clarity of the bands.

Moreover, we added the amount of proteins loaded in each lane in the figure legends.

Longer exposure did not solve the loading issue and beta-actin bands appeared saturated which definitely affect the protein normalization. The authors may replace with other representative images rather modify the current images, especially there 3 biological replicates and 3 technical replicates.

4 The labeling of figures “A, B, C…” was not indicated in the text.

We indicated in the test the panel labelling.

What I mean is to cite the figure in the text as Fig. 1A, Fig. 1B etc.

Author Response

I would like to thank the authors for responding to my initial comments, but I still have some concerns below.

  1. Line 90, the study was conducted only in male rats while hyperthyroidism is more common in

females than males. Some justifications are required. The number of animals used in each group

should be clearly stated.

We used rats of the same gender as an animal model of hyperthyroidism because we have long known that male and female rats have different responses to different pathophysiological conditions.

Venditti P, Piro MC, Artiaco G, Di Meo S. Effect of exercise on tissue antioxidant capacity and heart electrical properties in male and female rats. Eur J Appl Physiol Occup Physiol. 1996;74(4):322-9. doi: 10.1007/BF02226928. PMID: 8911824.

Di Meo S, de Martino Rosaroll P, Venditti P, Balestrieri M, De Leo T. Action potential configuration in heart papillary muscles from female rats in different thyroid states. Arch Physiol Biochem. 1997 Feb;105(1):58-65. doi: 10.1076/apab.105.1.58.13146. PMID: 9224547.

The aim of our study  was to understand whether ROS play a role in insulin resistance  induced by hyperthyroidism and not to investigate the heterogeneous response between genders that here we consciously avoided using only rats of the same gender.

The authors could include this as a limitation of the study in the discussion.

In the second revision we added a sentence in the conclusion underlying this aspect

  1. Figure 6A, the loading control beta-action are uneven for different groups and the Akt band for H+

group did not seem to develop well and the authors may consider choosing another representative

image. The amount of proteins loaded in each lane should be indicated in the figure legend.

In the present version of the manuscript, we used images for beta actin and Akt obtained after a longer period of exposure to improve the clarity of the bands.

Moreover, we added the amount of proteins loaded in each lane in the figure legends.

Longer exposure did not solve the loading issue and beta-actin bands appeared saturated which definitely affect the protein normalization. The authors may replace with other representative images rather modify the current images, especially there 3 biological replicates and 3 technical replicates.

We changed the images as suggested by the reviewer

4 The labeling of figures “A, B, C…” was not indicated in the text.

We indicated in the test the panel labelling.

What I mean is to cite the figure in the text as Fig. 1A, Fig. 1B etc.

We have corrected the name of the figures